# Stepped-wedge cluster-randomised controlled trial to assess the cardiovascular health effects of a managed aquifer recharge initiative to reduce drinking water salinity in southwest coastal Bangladesh: study design and rationale

Abu Mohd Naser,[1] Leanne Unicomb,[2] Solaiman Doza,[2] Kazi Matin Ahmed,[3] Mahbubur Rahman,[2] Mohammad Nasir Uddin,[2] Shamshad B Quraishi,[4] Shahjada Selim,[5] Mohammad Shamsudduha,[6] William Burgess,[7] Howard H Chang,[8] Matthew O Gribble,[1] Thomas F Clasen,[1] Stephen P Luby[9]

For numbered affiliations see end of article.

**Correspondence to**
Dr Abu Mohd Naser;
atitu@emory.edu

## ABSTRACT

**Introduction** Saltwater intrusion and salinisation have contributed to drinking water scarcity in many coastal regions globally, leading to dependence on alternative sources for water supply. In southwest coastal Bangladesh, communities have few options but to drink brackish groundwater which has been associated with high blood pressure among the adult population, and pre-eclampsia and gestational hypertension among pregnant women. Managed aquifer recharge (MAR), the purposeful recharge of surface water or rainwater to aquifers to bring hydrological equilibrium, is a potential solution for salinity problem in southwest coastal Bangladesh by creating a freshwater lens within the brackish aquifer. Our study aims to evaluate whether consumption of MAR water improves human health, particularly by reducing blood pressure among communities in coastal Bangladesh.

**Methods and analysis** The study employs a stepped-wedge cluster-randomised controlled community trial design in 16 communities over five monthly visits. During each visit, we will collect data on participants' source of drinking and cooking water and measure the salinity level and electrical conductivity of household stored water. At each visit, we will also measure the blood pressure of participants ≥20 years of age and pregnant women and collect urine samples for urinary sodium and protein measurements. We will use generalised linear mixed models to determine the association of access to MAR water on blood pressure of the participants.

**Ethics and dissemination** The study protocol has been reviewed and approved by the Institutional Review Boards of the International Centre for Diarrheal Disease Research, Bangladesh (icddr,b). Informed written consent will be taken from all the participants. This study is funded by Wellcome Trust, UK. The study findings will be disseminated to the government partners, at research conferences and in peer-reviewed journals.

### Strengths and limitations of this study

► This is the first study to evaluate the health impact of managed aquifer recharge (MAR) in southwest coastal Bangladesh.
► The stepped-wedge trial ensures we will have counterfactual data as well as gradual access to MAR water in all communities.
► Objective measurement of exposure (drinking water salinity) and outcomes (urinary sodium and blood pressure).
► The magnitude of exposure will vary geographically and across time period. Therefore, MAR water salinity will differ across communities at a single point of time, and also for the same community at different points of time.
► Compliance of the intervention may be different across sites and for individuals of different socioeconomic status.

**Trial registration number** NCT02746003; Pre-results.

## INTRODUCTION
### Background and rationale

Saltwater intrusion and salinisation have increased groundwater salinity in many coastal aquifers and small islands across the world.[1–5] This is driven by a number of climatological and anthropogenic factors including global warming, increased cyclones and tidal surges, reduced river discharge and increased groundwater abstraction in excess of recharge.[6–8] Communities in many

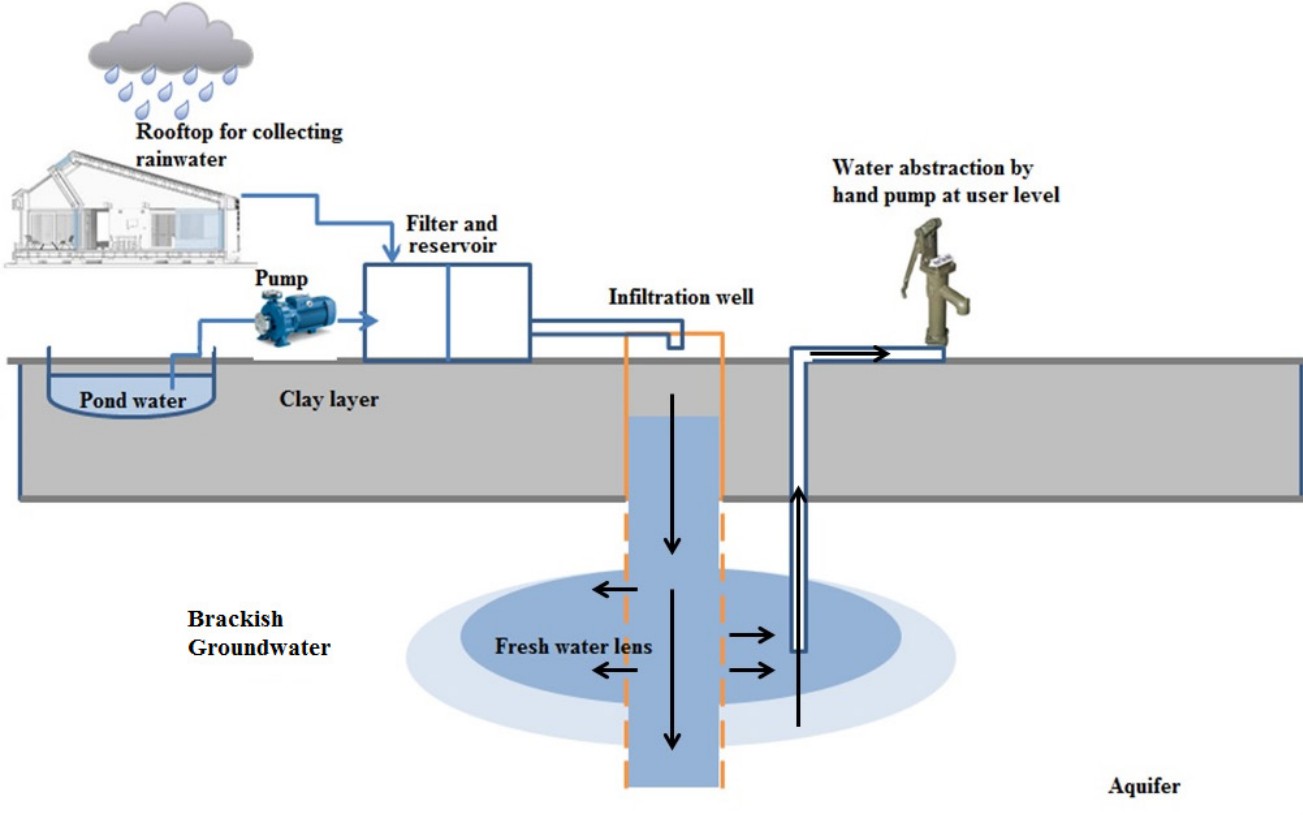

**Figure 1** Schematic diagram of the managed aquifer recharge systems in southwest coastal Bangladesh.

coastal regions rely on groundwater as their main source of drinking water[9] as well as freshwater for domestic, agricultural and industrial purposes.[10] Nearly half of the world's population resides in coastal areas[3] and 10% of these live in low-lying coastal areas where surface elevation is generally ≤10 m above mean sea level.[11] Salinisation in coastal areas is expected to increase in the future because of increased groundwater withdrawal due to population and economic growth and sea level rise.[3] As the world's population and economic activities continue

to grow, groundwater supplies are progressively under threat of depletion, which increases the importance of monitoring, management and conservation of coastal freshwater aquifers.[12 13]

One approach to minimise the impact of groundwater salinisation is to enhance groundwater recharge into coastal aquifers.[14] Managed aquifer recharge (MAR) is an approach to artificially promote freshwater recharge to increase storage. MAR involves infiltration of freshwater (eg, rainwater and pond water) into aquifers to

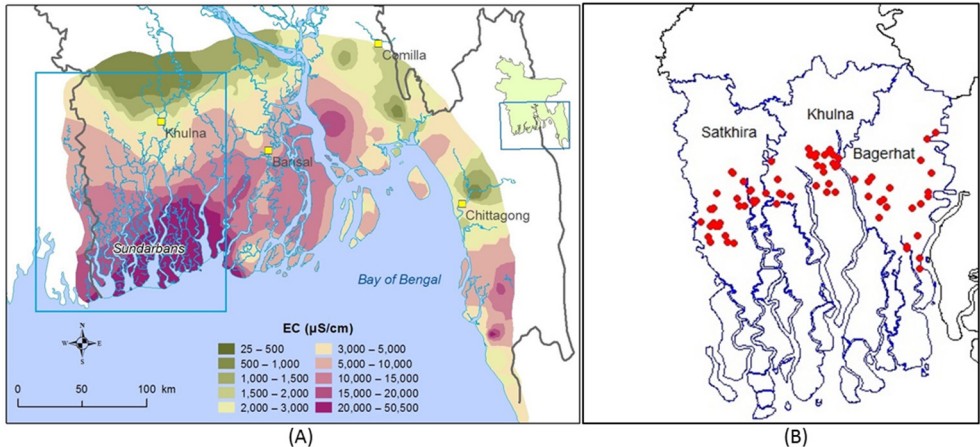

**Figure 2** (A) Groundwater salinity (measured as EC: electrical conductivity) distribution in coastal Bangladesh (data source: Bangladesh Water Development Board) and (B) location of the 75 managed aquifer recharge sites in southwest coastal region illustrated by red dots (data source: Professor Kazi Matin Uddin Ahmed).

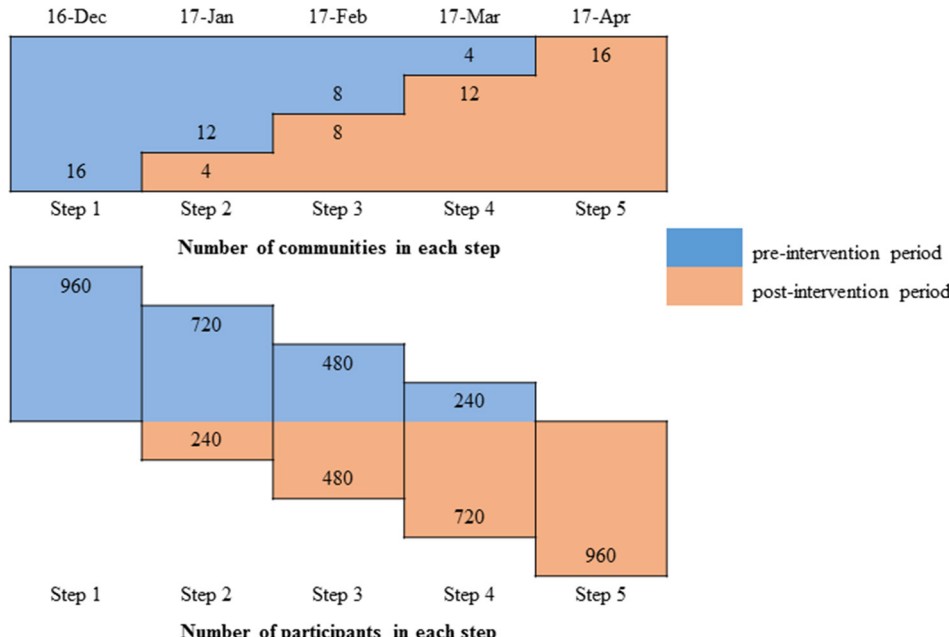

**Figure 3** Timeline of blood pressure measurement with number of managed aquifer recharge communities and participants in preintervention (blue colour) and postintervention (brown colour) time period in different steps.

create a store of freshwater within the naturally brackish aquifer (figure 1).[15–17] MAR represents a promising adaptive strategy for increasing freshwater availability and sustaining a year-round drinking water supply that is protected from evaporation, and could be resilient to tidal storms, cyclones and surface water salinity since freshwater infiltration and storage occur under confined conditions.[18]

The brackishness of groundwater in southwest coastal Bangladesh is caused by a combination of climatic and anthropogenic factors including sea level rise, frequent cyclones and tidal storms, and shrimp cultivation.[19–22] In the future, climate change and sea level rise are expected to cause more cyclones, tidal surges and flood in this region that will further affect surface water and groundwater salinity.[7] In many areas in southwest coastal Bangladesh, both shallow and deep aquifers contain naturally brackish water causing acute scarcity of drinking water (figure 2).[23 24] Water salinity in southwest coastal Bangladesh follows a clear seasonal pattern—higher in the dry season than the wet season.[25] Salinity of water bodies builds up from October to May, peaking during February to early May. After May, salinity decreases sharply due to rainfall and increased upstream river flow. People collect and use rainwater for drinking and cooking purposes during the monsoon when precipitation is intense, but during the dry season, they generally rely on pond water or saline tube well water.[26]

The Geology Department of the University of Dhaka, in collaboration with Unicef, and the Department of Public Health Engineering (DPHE), Government of Bangladesh, have piloted 20 small-scale MAR projects in three districts of southwest coastal Bangladesh to evaluate the feasibility of MAR for drinking water supply in rural communities.[27] The shallow brackish aquifer was the target storage zone that is overlain by Holocene clay aquitard of 3–15 m thickness.[16] In the pilot phase, an average storage of approximately $900 \, m^3$ of freshwater per year per site was established, sufficient to deliver 15 L of safe drinking water per day per household, which can fulfil the demand for drinking and cooking for approximately 300 people in 60–70 households during the dry season.[28 29] The second phase of the MAR project started in 2014 aiming to install 75 MAR systems in three districts of southwest coastal Bangladesh: Satkhira, Khulna and Bagerhat (figure 2B). In March 2016, thirty MAR sites were handed over to the community for water consumption. The remaining 45 sites are expected to be ready by November 2016 following infiltration of water during monsoon of 2016 (June to October 2016).

Epidemiological studies have demonstrated that high sodium intake is associated with elevated blood pressure[30–32] and other cardiovascular diseases.[33] A study conducted among the adult population residing in southwest coastal Bangladesh suggests drinking saline water was associated with high blood pressure after adjusting for personal, lifestyle and environmental factors.[34] Among study participants, the mean systolic blood pressure for those consuming water with sodium from the lowest quintile was 119.4 (SD 13.7) and from the highest sodium quintile was 126.7 mm Hg (18.0).[34] Excess sodium intake from drinking brackish water has been also associated with increased gestational hypertension and pre-eclampsia among pregnant women in southwest coastal Bangladesh.[35 36] Mean systolic and diastolic blood pressures were higher among the pregnant women who

drank tube well or pond (saline sources) water compared with those who drank rainwater.[35 37] The mean systolic blood pressure of pregnant women from these areas was 102.4mm Hg among those who drank rainwater, 112.6mm Hg among pond water users and 119.4mm Hg among brackish groundwater users.[35] High blood pressure during pregnancy is associated with high maternal mortality, and adverse pregnancy and fetal outcomes.[38]

There are 37million people living in the southwest coastal region and 20million are currently affected by drinking water salinity.[39] The estimated mean global sodium consumption is 3.95g/day (range 2.2–5.5g/day),[40] but people in southwest coastal Bangladesh consume up to 16g of sodium per day through drinking brackish groundwater.[41] While MAR water can potentially reduce exposure to saline water, the health effects of providing access to MAR water have not been investigated. Characterising the health consequences of shifting from current drinking water supply to MAR water can assist decision makers to assess the value of scaling up MAR. Many water interventions, for example, failed to achieve health impact, because people do not use them. There is also considerable controversy on whether reducing sodium intake improves health.[42] In addition, the MAR system will also alter the intake of other cations such as calcium and magnesium that may have health impact.[43–46] Because the cost of these systems is high and their long-term feasibility an open question, clarifying the amount of health benefit, if any, is an important step that can help inform future efforts to provide water to communities faced with high levels of groundwater salinity. The human subject committees who reviewed the study appreciated the uncertainty in bringing solutions to scale in this setting and considered it appropriate to include health measurements of participating residents to objectively evaluate the potential impact.

## Objectives
Randomised controlled trials demonstrate that reduction in dietary sodium in adults decreases blood pressure among people both with and without hypertension.[47–49] Meta-analysis of randomised trials also suggests modest reduction in salt intake for 4 or more weeks causes significant reduction in blood pressure at the population level.[50 51] The primary objective of the study is to assess whether access to low-salinity MAR water can reduce blood pressure of community members≥20 years of age. Secondary objectives include whether access to MAR water can reduce urinary sodium and total protein excretion. We will evaluate water salinity, urinary sodium excretion and blood pressure so that we understand whether or not we achieved our immediate targets along the causal pathway.

## METHODS
### Study design
We will implement a stepped-wedge cluster-randomised controlled trial in 16 MAR communities. MAR system is a community-based intervention designed to supply low-salinity water for 60–70 households in a village or community. MAR intervention cannot be implemented and randomised at individual or household levels. Once a community will have access to MAR water, it is difficult to withhold the access of MAR water for some households. Therefore, we will conduct a cluster-randomised trial where each community will be considered as a cluster. The stepped-wedge design allows communities to gradually have access to MAR water; however, the point at which their access commences will be randomly assigned.[52] In this way, each MAR site will contribute data for both the intervention and the control time periods. We will have five monthly steps in the stepped-wedge trial (figure 3). In the first step, none of the communities will have MAR water available for drinking and we will collect baseline information. During each subsequent month, four randomly selected communities will receive access to MAR water for drinking and cooking. In the last (fifth) monthly step, all the communities will have access to MAR drinking water.

### Study settings
The study will be conducted at the community level in three districts in southwest coastal Bangladesh—Khulna, Satkhira and Bagerhat—where 75 MAR systems have been installed. Of these, 30 systems are already in use. We will select 16 communities for study based on consultation with the implementers of MAR systems (Dhaka University and Unicef Bangladesh). Three criteria need to be met for inclusion of communities in the study: communities have not started drinking MAR water, acceptable level of arsenic (<50μg/L) in drinking water set by the government of Bangladesh[53 54] and electrical conductivity (a measure of salinity) of water below 2000 μS/cm, an indication that the MAR system has successfully reduced salinity in the aquifer.

### Participant eligibility criteria
The inclusion criteria will be household members ≥20 years old and pregnant women who willingly agree to exclusively use MAR water during the dry season for drinking and cooking purposes. The Dhaka University team have determined the catchment area for each MAR site based on geographical distance from MAR infrastructure and have developed a list of households who expressed willingness to drink MAR water. Any ≥20-year-old hypertensive household members will be also eligible for participation but research staff will collect information about their medication if any. Research staff will enrol postadolescent ≥20-year population as an individual's response to salt intake (salt-sensitivity) increases with age[55] and adolescence is associated with adrenal and nervous system maturation that may contribute to salt sensitivity.[56 57] They will approach households living near the MAR water access point. They will enrol the 28 closest households surrounding each MAR system that meet the inclusion

criteria and consent (online supplementary information 1) to study participation. During the informed written consent process, research staff will inform each household that they will be randomly selected when to start consuming MAR water as part of the scientific process of the study. All household members >20 years of age will be eligible and enrolled in the study from the selected 28 households. In addition, research staff will enrol households that include a pregnant woman in each selected MAR catchment community irrespective of household selection from a list of pregnant women developed by a female promoter.

## Randomisation and intervention delivery

We will randomly select four MAR communities to drink MAR water in each step following the first (baseline) step (figure 3) by computer-generated random numbers. Randomisation will be conducted by an investigator who will not be directly involved in implementation of the stepped-wedge study and this will be done before commencement of the study. The study could not be blinded, therefore there was no concealment from the cluster participants. Although infiltration of rainwater and pond water into the brackish aquifer through the MAR systems will be ongoing for 1–2 years in each community, people will not have access to MAR water until a formal handover of MAR systems to community members. The implementers hire a caretaker or gatekeeper for each MAR community who is responsible for maintenance of the respective MAR system prior to the community handover. The implementers form a community management team who are responsible for maintenance of each MAR system following handover. We have synchronised the community handover with the randomisation schedule in the 16 communities after consulting with the gatekeepers. Community health promoters will inform the participants when the MAR water will be available for consumption once the community management team is formed. Agreement of the implementers and gatekeepers was sought at the beginning of the study during site selection for inclusion of any site in trial and access of MAR water as per the randomisation.

We will deploy a local trained promoter at each MAR site, who will visit these households, list members and identify pregnant women in these and other households in catchment areas of each MAR site. Promoters will encourage all household members to drink MAR water exclusively while they are at home, carry a bottle of MAR water while they go out for work and other activities, and to cook with MAR water. The community health promoters will visit households with promotional materials (eg, flip charts) from the beginning of the study to inform household members about adverse health effects of drinking brackish water and potential benefits of drinking low-salinity MAR water. As per the randomisation schedule, they will inform households when MAR water will be available for consumption.

## Sample size

The sample size of the stepped-wedge trial was calculated based on the effect of MAR systems to reduce systolic blood pressure of the ≥20-year-old community members. Data from southwest coastal Bangladesh demonstrated a 7 mm Hg lower mean systolic blood pressure among pregnant women who drank pond water compared with those who drank brackish water, and a 17 mm Hg lower mean systolic blood pressure for pregnant women who drank rainwater compared with those who drank brackish water.[35] The mean reduction in systolic blood pressure following long-term modest salt reduction is 4.2 mm Hg.[50] For calculations, we considered a mean systolic blood pressure reduction of at least 3 mm Hg among the communities who will have access to MAR water for drinking and cooking purposes compared with those who will use brackish water. The entire population under the MAR community catchment area will be considered as one cluster. Each MAR community serves approximately 300 people, constituting 60–70 households in the catchment area.[28] The mean household size in southwest coastal Bangladesh is 4.2, approximately 52% of household members are ≥20 years old.[58 59] Since we will collect blood pressure from all household members ≥20 years old, we estimated an average of 2.2 participants in this age group per household. We will select 28 households per cluster (28×2.2=60 people per cluster). A previously published study in Bangladesh reported SD 13.51 for systolic blood pressure,[60] but we considered SD 20 for systolic blood pressure to capture a greater variation of blood pressure. We calculated the sample size for the stepped-wedge trial adopting the formula used by Hussey and Hughes.[61] Although their formula assumed no within-participant correlation over time (cross-sectional design), we applied their formula for a cohort design (same participant followed up over time). The salinity problem and the MAR water quality may be influenced by seasonality over the progression of dry season and participants' blood pressure may respond differently to different level of sodium exposure in MAR drinking water. Therefore, it is important that we investigate the effect of the MAR system on the same participants over the entire dry season. Since we will investigate the same participants in each step, we will have less participant-level variability and sufficient power compared with a cross-sectional design to investigate the effect of MAR systems. We initially calculated the sample size for a simple unadjusted trial ($N_U$), and then multiplied by the design effect for a stepped-wedge trial ($DE_{SW}$). The sample size for a simple unadjusted trial was calculated as 1396 participants (effect size=3, SD of 20 for both groups, 80% power and 5% type I error). The $DE_{SW}$ was 3.04 based on a formula provided by Hussy and Hughes, therefore sample size was 1396×3.04. We also inflated the total sample size considering 10% loss to follow-up. We then followed the approaches of determining the number of clusters required for total sample size

 

considering a fixed cluster size of 60.[62] We then calculated the number of steps that we need to randomise by dividing the total cluster by steps. We calculated that 16 MAR communities will be required for the study with four communities randomised to access MAR water in each step (figure 3).

## Data collection methods

The outcomes of interest will be measured from selected participants of 16 communities irrespective of their access to MAR water during five monthly visits. The interval between two successive blood pressure measurements or two consecutive data collection visits will be at least 4 weeks for each participant. After taking informed written consent from household heads and each participant, research staff will administer a survey using a structured questionnaire to collect information on drinking water sources, household members' sociodemographic status, medical history, family history of cardiovascular diseases, medication intake, and cardiovascular risk factors including smoking history and alcohol intake during the first monthly visit. Exposure and outcome data will be collected in all five visits. Participants' weight and hip circumference will be measured in all visits but height will be measured in one visit.

As part of the MAR uptake evaluation, the research staff will collect self-reported data on MAR water use from each household and whether households exclusively or intermittently use MAR water for drinking and cooking during follow-up visits. Research staff will also collect stored drinking and cooking water samples for measurement of electrical conductivity and will ask respondents the source of stored water.

## Exposure assessment

During each visit, research staff will inquire about household members' reported primary water sources used for drinking and cooking purposes in the last 24 hours, collect reported information on whether participants exclusively used the primary water sources and explore whether any alternative water sources were used. They will also ask about the frequency of collection and cost of primary water sources for drinking and cooking, time required to collect water, amount of collected water and when the last water was collected. Research staff will observe the presence of stored water in households, ask the sources of each container of stored water and will collect water samples that have been stored for drinking and cooking (if any) to measure the salinity level, electrical conductivity, resistivity, total dissolve solutes and temperature using Hanna® Salinity meter. They will also collect the MAR water from the source MAR outlet to measure the salinity level, electrical conductivity, resistivity, total dissolved solutes and temperature. Research staff will collect 24 hours urine from each participant to measure the urinary sodium excretion as a proxy for sodium intake.[63 64]

## Outcomes assessment

Systolic blood pressure of the ≥20-year-old participants is the primary outcome of this study. Secondary outcomes will be diastolic blood pressure, mean arterial pressure and pulse pressure (table 1). The different components of blood pressure are independent cardiovascular risk factors associated with sodium intake,[65–69] and indicate future risk for different cardiovascular diseases.[70] High systolic blood pressure puts more stress on the vascular wall and is strongly associated with risk for future intracerebral haemorrhage, subarachnoid haemorrhage, angina, myocardial infarction and peripheral vascular diseases.[70] Raised diastolic blood pressure is associated with aortic and thoracic aneurysm.[70] Some effects of high sodium intake are independent of high systolic and diastolic blood pressures such as arterial stiffness and left ventricular mass—both are independent predictors of future cardiovascular diseases. High pulse pressure (difference between systolic and diastolic blood pressures) is associated with increased arterial stiffness,[66 68] and high mean arterial pressure (diastolic blood pressure plus one-third

**Table 1** Study outcomes including normal ranges of blood pressure and biomarkers

| Outcomes | Study population | | Measures | Normal range/calculation |
| | ≥20 years old | Pregnant women | | |
| --- | --- | --- | --- | --- |
| Primary outcome: systolic blood pressure | Yes | Yes | Systolic blood pressure | 90–140 mm Hg |
| Secondary outcomes: diastolic blood pressure | Yes | Yes | Diastolic blood pressure | 60–90 mm Hg |
| Mean arterial pressure | Yes | Yes | Mean arterial pressure | Diastolic blood pressure + one-third systolic blood pressure |
| Pulse pressure | Yes | Yes | Pulse pressure | Systolic blood pressure – diastolic blood pressure |
| Tertiary outcomes: urinary sodium excretion | Yes | Yes | Urinary sodium-creatinine ratio | 40–220 mEq/L/24 hours |
| Proteinuria | Yes | Yes | Urinary protein-creatinine ratio | <0.11 mg/mg |

of systolic blood pressure) and high pulse pressure are independently associated with increased left ventricular mass.[66 68]

We will also measure the creatinine adjusted protein excretion of all participants as tertiary outcomes (table 1). Proteinuria is a biomarker for future risk of cardiovascular diseases,[71–74] and is associated with the pathogenesis of cardiovascular diseases, including hypertension,[75 76] chronic kidney disease,[77] myocardial ischaemia,[78] carotid artery thickness,[79 80] left ventricular hypertrophy,[81 82] hyperlipidaemia,[83] atherosclerosis[84] and coronary artery calcification.[72 80 85 86] Reduced salt intake for 4 weeks has been associated with decreased proteinuria in blinded randomised controlled trial.[87] Pre-eclampsia is associated with high maternal mortality and adverse pregnancy and fetal outcomes[38] and is characterised by high blood pressure and proteinuria after 20 weeks of pregnancy.[88]

### Blood pressure measurement

Blood pressure will be measured for all ≥20-year-old participants and pregnant women during each step. Research staff will use an Omron HEM-907 blood pressure device, which is comparable to the gold standard mercury sphygmomanometers in terms of measurement and meets the Association for the Advancement of Medical Instrumentation's criteria.[89] Blood pressure will be measured following the guidelines described by Pickering *et al* and Giorgini *et al*.[90 91] Caffeine (tea, coffee, carbonated beverages), eating, heavy physical activities and smoking will be proscribed for 30 min prior to measuring blood pressure. Participants will rest for 5 min sitting on a chair keeping their arm supported. An appropriate size cuff and calibrated instrument will be used for different age groups and the blood pressure instrument will be positioned at heart level. Blood pressure will be measured three times; first left arm, then right arm, then again left arm. Both systolic and diastolic blood pressures will be recorded from all measurements. The arithmetic mean of three systolic blood pressure measurements will be used as the primary outcome. However, if a systolic blood pressure measurement differs by 10% from the other measurements, that measurement will be excluded when calculating the arithmetic mean systolic blood pressure.

### Biomarker measurements

Field research staff will instruct the participants to collect 24 hours urine sample during each household visit. The volume of the urine samples will be noted at household level, and a sample of 25 mL urine will be collected and transported to the field laboratory at 2–8°C within 6 hours of collection for processing, analysis and storage. Aliquots of each participant's urine sample will be made for biochemical and electrolyte measurements. Urinary creatinine concentration will be measured by a colorimetric method (Jaffe reaction) using a semiauto biochemistry analyser (Evolution 3000, BSI, Italy). Urinary total protein will be estimated using a light sensitive coloured reagent (Randox, UK). We will use the

direct ion selective electrode method for urinary sodium measurements using a semiauto electrolyte analyser (Biolyte2000, Bio-care, Taiwan). We will measure the uric acid concentration in blood by an enzymatic colorimetric method (Evolution 3000, BSI). We will perform routine quality control for all tests using standard quality control reagents (Bio-Rad Laboratories, USA) and 5% of samples will be cross-checked at the International Centre for Diarrhoeal Disease Research, Bangladesh (icddr,b) central laboratory in Dhaka.

### Data management and monitoring

The data collection instrument will be programmed and field research staff will use handheld computers to collect data. Appropriate range values and data values will be programmed to minimise data entry errors. The data set downloaded from handheld devices will be cleaned and checked by the site investigators. All laboratory data will be double entered. Data will be stored in icddr,b's data repository system, in compliance with the system's requirements and will be publicly available after analysing the primary result.

The research staff will be trained for identifying adverse events such as hypertension and hypertensive disorders in pregnancy. They will report to the investigators following identification of these patients and the investigators will assess whether these adverse events need to be reported to icddr,b's Ethical Review Committee.

### Statistical methods

We will conduct an intention-to-treat analysis for the primary analysis. For the primary analysis, we will assess whether access to MAR water reduces the systolic blood pressure (continuous outcome) of the ≥20-year-old participants. Pregnant women will be included in the primary analysis because it is likely few pregnant women will be identified in the 16 communities and separate analysis of pregnant women will be underpowered. We will use generalised linear mixed models with appropriate links for the primary analysis considering random effects for community, households and participants, and fixed effects of steps or visits. We will adjust the effect of MAR systems on blood pressure for age, sex, weight and height, and personal and socioeconomic factors (table 2). We will inspect the missing data patterns and use multiple imputation with chained equations to jointly impute data on missing exposure and confounders to preserve an unbiased association estimate if the data are missing at random conditional on measured variables. We will also conduct subgroup analyses among the households that adhere with the MAR intervention and that exclusively use MAR water. If the trial demonstrates drinking MAR water has health effects, we will investigate the time required for health effects following receiving MAR intervention in secondary analysis.

**Table 2** Key variables to be used for assessing the health impact of access to MAR water

| Outcomes | Exposures | Confounders | Covariates | Moderators |
|---|---|---|---|---|
| Systolic blood pressure | Access to MAR water | Sex | Age | Drinking/cooking from another water sources |
| Diastolic blood pressure | Drinking water salinity | Socioeconomic status | Weight | |
| Pulse pressure | Cooking water salinity | Education | Height | |
| Mean arterial pressure | Drinking water sources Cooking water sources | | Waist circumference | |
| Urinary total protein Urinary sodium excretion | | | Mid-upper arm circumference | |
| | | | Adding table salt | |
| | | | Adding salt for cooking | |
| | | | Exercise level | |
| | | | Smoking status | |
| | | | Use of betel nuts | |
| | | | Stress | |
| | | | Diet | |
| | | | Urinary creatinine | |

MAR, managed aquifer recharge.

## Ethics

Informed written consent will be taken from all participants and household heads. Consent will be also taken for ancillary studies and future use of specimens collected from study participants. This study protocol has been reviewed and approved by the icddr,b Ethical Review Committee. Approval will be taken for any addition or modification of the protocol from icddr,b Ethical Review Committee. If research staff members identify patients with hypertension or cases of hypertension during pregnancy, they will refer patients to the local government health facilities for further management. In addition, research staff will train pregnant women and family members to recognise the danger signs during pregnancy, and will also instruct them where to seek medical care if such danger signs appear. During monthly household visits, research staff will encourage pregnant women to attend prenatal visits. All data sets will be anonymous without the personal identifiers and participants' privacy will be maintained during data storage, analysis and dissemination.

## Dissemination

Study findings will be shared with the DPHE, Department of Environment of the government of Bangladesh and other partner non-governmental organisations working in southwest coastal Bangladesh for access to safe drinking water. We will discuss the scope and limitations of the MAR system to address the demand of safe drinking water based on findings from this study. We will be submitting abstracts to international conferences for dissemination to the international audience working on safe water. We will develop manuscripts and submit to peer-reviewed journals to publish our research.

## DISCUSSION

This will be the first study to assess the health impact of an environmental intervention to reduce groundwater salinity in southwest coastal Bangladesh. Our study has several strengths. The stepped-wedge trial ensures that we will have counterfactual data as well as gradual access to MAR water in all communities. In a stepped-wedge design, treatment effect of an intervention can be estimated from between and within-cluster comparisons as participants will act as their own control, compared with only between-cluster comparison in a parallel cluster-randomised design.[61]

We explored other study designs such as non-randomised study. As per suggestion of icddr,b's Ethical Review Committee, we conducted a pilot study in four communities (two MAR communities and two non-MAR communities) from May to August 2016. Pilot study findings suggested that because of the heterogeneity in village-level hydrogeological conditions and in participation by community residents with different characteristics, non-randomised quasiexperimental studies would be expected to generate unequal distribution of confounders that would undermine scientific inference. The objective of the pilot study was to pretest our data collection instrument laboratory protocols prior commencement of the main study. We identified unequal distributions of several important observed confounders across the communities such as socioeconomic status, and household characteristics such as supply of electricity and ownership refrigerator. We identified that some covariates such as supply of electricity were perfectly predictive of communities receiving MAR interventions. We were unable to derive satisfactory matching of MAR intervention and control households after principal component analysis. The findings highlighted the importance of a randomised

trial to evaluate the health effects of MAR systems since there are many observed and unobserved household and community-level and hydrogeological confounders that may influence blood pressure.

We also explored the possibilities of parallel group randomised trial but several programmatic issues barred us to implement a parallel group cluster-randomised trial. The implementers of MAR systems (Unicef Bangladesh and University of Dhaka) installed the 75 new sites in different stages and planned to allow communities to access these over three dry seasons. The implementers informed us that 20–25 MAR sites may be suitable for community access during our study period and they wanted to allow all communities to drink MAR water in the same season after satisfactory reduction in salinity. Therefore, a parallel group cluster-randomised trial was not suitable as we could not randomly select control sites and postpone the community to drink water for the dry season. We believe that the stepped-wedge trial design was appropriate to obtain counterfactual data as well as fulfilling the programmatic need.

We will use the same instruments for data collection and outcome measurements for all steps for exposed and unexposed communities, which will mitigate bias in data collection. Measurement of water source salinity and urinary sodium concentration will explain the biologically plausible effect of drinking MAR water on blood pressure. All outcomes planned to measure in the study are objective outcomes that will reduce the risk of reporting bias. Collection of detailed exposure and covariate data will help in determining a valid association between drinking MAR water and health benefits. We will have several outcome variables and biomarkers that will ensure a comprehensive health benefit evaluation of access to MAR water.

The study has several limitations. We will be unable to control salinity level of drinking water across the 16 MAR communities due to different hydrogeological conditions and volume of infiltrated water across the aquifers. Although the initial electrical conductivity of MAR water will be <2000 μS/cm, electrical conductivity is likely to vary across different MAR communities with the intensification of dry season. Therefore, MAR water salinity will differ across communities at a single point of time, and also for the same community at different points of time. This will result in different versions of exposures (ie, access to MAR water with different salinity levels) that may affect different responses for some participants depending on the version of exposure they will receive.[92 93] To account this, we will specify the versions of exposure by measuring the salinity level of MAR water available in participants' households and interpret the response as unit change of salinity at different MAR water salinity levels. As compliance of drinking MAR water may be different across communities despite active encouragement by promoters, one problem of the intention-to-treat analysis is that if the proportions of participants who will always drink MAR water are low compared with those who will not, the potentially greater effect of MAR intervention on blood pressure of fully participating individuals may be washed out by the smaller effect of those who will not comply. To account for this, as a secondary analysis, we will conduct 'instrumental variable' (IV) analyses by jointly running two regression models: a regression model for predicting urinary sodium excretion by drinking water salinity, and a regression model for blood pressure prediction given the urinary sodium excretion. The assumption of IV analyses is fulfilled in our context as the exogenous variable (IV) can only affect outcome variable (blood pressure) through influencing the endogenous variable (urinary sodium excretion).[94] If MAR water has lower salinity than other water sources and if participants always drink MAR water, their urinary sodium excretion will be low compared with those who will not fully participate.

### Author affiliations
[1]Department of Environmental Health Sciences, Rollins School of Public Health, Emory University, Atlanta, Georgia, USA
[2]Environmental Health & Interventions Unit, Enteric and Respiratory Infections Program, Infectious Disease Division, International Centre for Diarrhoeal Disease Research, Dhaka, Bangladesh
[3]Department of Geology, Dhaka University, Dhaka, Bangladesh
[4]Analytical Chemistry Laboratory, Atomic Energy Centre, Bangladesh Atomic EnergyCommission, Dhaka, Bangladesh
[5]Department of Endocrinology & Metabolism, Bangabandhu Sheikh Mujib Medical University, Dhaka, Bangladesh
[6]Institute for Risk and Disaster Reduction, Departmentof Geography, University College London, London, UK
[7]Department of Earth Sciences, University College London, London, UK
[8]Department of Biostatistics and Bioinformatics, Rollins School of Public Health, Emory University, Atlanta, Georgia, USA
[9]Stanford Woods Institute for the Environment & Freeman Spogli Institute for International Studies, Stanford University, Stanford, California, USA

**Acknowledgements** The authors would like to thank Dr Karla Hemming, Institute of Applied Health Research, University of Birmingham, for her input in design and sample size calculation. The authors are grateful to the Dhaka University/Unicef field team for their assistance in providing preliminary data about the readiness of MAR sites.

**Contributors** SPL and AMN developed the study concept. AMN, SPL, LU, KMA and SD have developed the study design. MR, MOG, TFC and HHC reviewed the epidemiological study design. KMA, MS, WB, SS and MNU provided input in environmental and biological sample collection and analysis. HHC and MOG provided input in statistical analysis. AMN drafted and all authors reviewed the manuscript.

**Funding** The study has been funded by Wellcome Trust, UK (Grant no. 106871/Z/15/Z), Our Planet, Our Health Award. Wellcome Trust reviewed and approved the design as the condition of providing funding but will not have a role in data collection, data analysis, data interpretation or writing of the report. The study was implemented by International Centre for Diarrhoeal Disease Research, Bangladesh (icddr,b). icddr,b acknowledges with gratitude the commitment of Wellcome Trust, UK, to its research efforts. icddr,b is also grateful to the Governments of Bangladesh, Canada, Sweden and the UK for providing core/unrestricted support. MOG is supported by funding from the National Institute for Environmental Health Sciences to the Emory Health and Research Exposome Center (P30 ES019776).

**Competing interests** None declared.

**Patient consent** Detail has been removed from this case description/these case descriptions to ensure anonymity. The editors and reviewers have seen the detailed information available and are satisfied that the information backs up the case the authors are making.

**Ethics approval** ERC of International Centre for Diarrhoeal Disease Research, Bangladesh (icddr,b).

**Provenance and peer review** Not commissioned; externally peer reviewed.

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
