## [Reviewer comments · BMJ Open]

ARTICLE DETAILS

TITLE (PROVISIONAL)	Stepped-wedge cluster-randomized controlled trial to assess the cardiovascular health effects of a managed aquifer recharge initiative to reduce drinking water salinity in southwest coastal Bangladesh: study design and rationale
AUTHORS	Naser, Abu; Unicomb, Leanne; Doza, Solaiman; Ahmed, Kazi; Rahman, Mahbubur; Uddin, Mohammad; Quraishi, Shamshad; Selim, Shahjada; Shamsudduha, Mohammad; Burgess, William; Chang, Howard; Gribble, Matthew; Clasen, Thomas; Luby, Stephen

VERSION 1 - REVIEW

REVIEWER	Warren C. Jochem Dept. of Geography and Environment University of Southampton United Kingdom
REVIEW RETURNED	06-Mar-2017

GENERAL COMMENTS	Thank you for the opportunity to review this protocol. This study will examine the effects of a program to reduce exposure to saline water in southwest Bangladesh. Exposure to high salinity levels is a major problem in coastal Bangladesh and in many areas around the world, making this an important study. The protocol describes a stepped wedge trial (SWT) to evaluate a managed aquifer recharge (MAR) program. The analyses are based on intent-to-treat using generalized estimating equations (GEE), though the authors also mention a "random effect model" of outcomes for pregnant women and instrument variable analyses as possible alternatives. My main comment is that at times, the aim of the study is not clear. Is the aim to assess exposure to lower levels of water salinity for a measureable drop in blood pressure? Or is the aim to assess the implementation of a community-based intervention project of MAR as measured through blood pressure changes? For the first example of this confusion, see the abstract (page 2 lines 22-35). While the difference in these two aims may seem small, it raises several issues stemming from where is the uncertainty that is being studied? Is it uncertain if lower salinity intake can reduce blood pressure? Is it uncertain if MAR systems adequately reduce salinity levels in the water? Or is it uncertain if MAR systems can have broad enough uptake in different communities to effectively reduce exposure and subsequently blood pressure? The study protocol describes requirements that a MAR system already produce safe water and it only selects households who are willing to commit to using it exclusively (page 8). Moreover, if these MAR systems are already installed (as of fall
---

	2016), they are known to reduce salinity levels, lower sodium intake has health benefits, and people want to use them (page 8, lines 33-43; page 9, lines 15-23), what is the ethical justification for delaying access to the MAR water for up to 4 months for some communities? The protocol only gives analytical justification of obtaining counterfactuals for using the SWT. Did the project consider other study designs? Other comments: Page 8, line 38 – what does this study consider to be an “acceptable level” of arsenic? Page 8, line 45 – does the study exclude from participation people who already have a diagnosis of high blood pressure or hypertension? Does the study collect information on participants’ medications that could affect blood pressure? Page 9, line 11 – how will the analyses handle if a woman in one of the selected participant households becomes pregnant during the study? Page 9, line 25 – please provide more information on the methods the trained promoter will use to encourage use of MAR. If the aim of the study is to evaluate uptake/use of the intervention, there needs to be discussion on the information campaigns and promotion of MAR water. Also, how will communities who are not selected to receive the intervention until later phases be motivated to continue participating in the study? GEE are sensitive to missing data. Please explain more about the multiple imputation methods you will consider and how you will evaluate if they are sufficient (page 14 line 46). Page 15, line 3 – please provide more details on the “random effect model.” This term can mean different things across disciplines. What sample size will be needed? Is it likely you will recruit enough pregnant women? Figure 2 – please list the sources of the data used in each map. With these clarifications and suggestions noted above, I feel that publishing this protocol will make a good contribution to the literature and I look forward to reading the results of the study in the future.
--	--

REVIEWER	Karla Hemming Uni Bham uk
REVIEW RETURNED	28-Apr-2017

GENERAL COMMENTS	Thank you for asking me to review this paper. It is well written but a few items need to better reported, these are as follows: Will you adjust for time effects? This is crucial in this sort of study When people are enrolled will they be told when their cluster will transition? What sort of information are they told when participants are enrolled into the study? Sample size – please be clear that the 1396 is the sample size needed under individual randomisation. The sample size calculation is based on the Hussey and Hughes
---

	method which assumes different people are measured at each assessment time. If the TSS needed under individual randomisation is 1396×3, then the number of people per cluster needed is $1296 \times 3 / 16$ which is in the region of 250 – these should be divided between the number of measurement points. The authors have applied a sample size formula for a cross sectional design to a cohort design. Some acknowledgement of this is needed – it is likely that this means that authors calculation is conservative. A statistical methods section is needed. How the data will be analysed must be given due consideration and be reported at this stage. Likewise, information on sub-group effects, sensitivity analyses should also be reported. This is a minor point, and almost obvious, but it is usually standard to report a justification for using a cluster design. I might have missed this, but a schematic representation of the design might be useful. Details on how the randomisation will be conducted, by whom, how it will be concealed from the clusters and when the clusters will be informed of their switch date. Who recruited the individual participants.
--	--

VERSION 1 – AUTHOR RESPONSE

Reviewer: 1

Reviewer Name: Warren C. Jochem

Institution and Country: Dept. of Geography and Environment, University of Southampton, United Kingdom

Please state any competing interests or state 'None declared': None declared

Please leave your comments for the authors below

Comment- Thank you for the opportunity to review this protocol. This study will examine the effects of a program to reduce exposure to saline water in southwest Bangladesh. Exposure to high salinity levels is a major problem in coastal Bangladesh and in many areas around the world, making this an important study. The protocol describes a stepped wedge trial (SWT) to evaluate a managed aquifer recharge (MAR) program. The analyses are based on intent-to-treat using generalized estimating equations (GEE), though the authors also mention a “random effect model” of outcomes for pregnant women and instrument variable analyses as possible alternatives.

My main comment is that at times, the aim of the study is not clear. Is the aim to assess exposure to lower levels of water salinity for a measurable drop in blood pressure? Or is the aim to assess the implementation of a community-based intervention project of MAR as measured through blood pressure changes? For the first example of this confusion, see the abstract (page 2 lines 22-35).

While the difference in these two aims may seem small, it raises several issues stemming from where is the uncertainty that is being studied? Is it uncertain if lower salinity intake can reduce blood pressure? Is it uncertain if MAR systems adequately reduce salinity levels in the water? Or is it uncertain if MAR systems can have broad enough uptake in different communities to effectively reduce exposure and subsequently blood pressure? The study protocol describes requirements that a MAR system already produce safe water and it only selects households who are willing to commit to using it exclusively (page 8).

Moreover, if these MAR systems are already installed (as of fall 2016), they are known to reduce salinity levels, lower sodium intake has health benefits, and people want to use them (page 8, lines

33-43; page 9, lines 15-23), what is the ethical justification for delaying access to the MAR water for up to 4 months for some communities?

Response- Thank you very much for this comment. This is an evaluation of a proposed public health intervention that has only been implemented in a few dozen communities. The objective of the study is to assess whether access to low-salinity MAR water can reduce the blood pressure of community members ≥ 20 years of age. We are evaluating water salinity and urinary sodium excretion so that we understand whether or not we achieved our immediate targets along the causal pathway. Many water interventions, for example, failed to achieve health impact, because people don't use them. There is also considerable controversy on whether reducing sodium intake improves health¹. In addition, the MAR system will also alter the concentration of other cations that may have the health impact. Because the cost of these systems is high and their long-term feasibility an open question, clarifying the amount of health benefit, if any, is an important step that can help inform future efforts to provide water to communities faced with high levels of groundwater salinity. The human subjects committees who reviewed the study appreciated the uncertainty in bringing solutions to scale in this setting and considered it appropriate to include health measurements of participating residents to objectively evaluate the potential impact. We have strengthened the rationale section of the revised manuscript to clarify this.

Comment- The protocol only gives analytical justification of obtaining counterfactuals for using the SWT. Did the project consider other study designs?

Responses- We explored other study designs such as non-randomized study and parallel group cluster randomized trial. As per suggestion of icddr,b's IRB, we conducted a pilot study in four communities (two MAR communities and two non-MAR communities) during May-August 2016. Pilot study findings suggested that because of the heterogeneity in village level hydrogeological conditions and in participation by community residents with different characteristics, non-randomized quasi-experimental studies would expect to generate an unequal distribution of confounders that would undermine scientific inference. The objective of the pilot study was to pre-test our data collection instruments laboratory protocols prior commencement of the stepped-wedge trial. We identified unequal distributions of several important observed confounders across the communities such as socioeconomic status and household characteristics such as the supply of electricity and ownership refrigerator. We identified that some covariates such as the supply of electricity were perfectly predictive of communities receiving MAR interventions. We were unable to derive satisfactory matching of MAR intervention and control households after principal component analysis. The findings highlighted the importance of a randomized trial to evaluate the health effects of MAR systems since there are many observed and unobserved household and community-level and hydrogeological confounders that may influence blood pressure.

We also explored the possibilities of parallel group randomized trial but several programmatic issues barred us from implementing a parallel group cluster randomized trial. The implementers of MAR systems (UNICEF Bangladesh and University of Dhaka) installed the 75 new sites in different stages and planned to allow communities to access these over three dry seasons. The implementers informed us that 20-25 MAR sites may be suitable for community access during our study period and they wanted to allow all communities to drink MAR water in the same season after satisfactory reduction in salinity. Therefore, a parallel group cluster randomized trial was not suitable as we could not randomly select control sites and postpone the community to drink water for the dry season. We believe that the SWT design was appropriate to obtain counterfactual data as well as fulfilling the programmatic need.

We have included these paragraphs in the discussion section of the revised manuscripts.

Other comments:

Page 8, line 38 – what does this study consider to be an “acceptable level” of arsenic?

Response- We followed the Government of Bangladesh “acceptable level” of arsenic in drinking water for Bangladesh is $<50 \mu\text{g/L}$. We have added this limit in the revised manuscript. Acceptable arsenic concentration in water is one of the eligibility criteria of aquifer selection for MAR installation. The implementers have a routine monitoring systems to assess the arsenic concentration in MAR water after initiation of infiltrating water into the aquifer and they don't allow communities to drink MAR water if the arsenic concentration is above the Bangladesh standard.

Comment- Page 8, line 45 – does the study exclude from participation people who already have a diagnosis of high blood pressure or hypertension? Does the study collect information on participants' medications that could affect blood pressure?

Response- We did not exclude participants with hypertension from the study. We're collecting information on participants' medication that could affect blood pressure. The pilot study suggested many people are undiagnosed with hypertension or do not adhere to the medications in rural Bangladesh. We will include participants irrespective of hypertension status in the primary analysis.

Comment- Page 9, line 11 – how will the analyses handle if a woman in one of the selected participant households becomes pregnant during the study?

Responses- If any enrolled women become pregnant during the study, we will collect her blood pressure and urine protein data during the monthly visits. We will also refer the pregnant women for antenatal care in the government health facilities. Since pregnant women's physiology is different than a normal adult, we will conduct separate analysis for pregnant women. We will conduct linear mixed models to assess whether access to water from MAR systems reduce blood pressure and urinary protein among pregnant women. We'll consider the random effects of communities and pregnant women to determine the health effects of access to MAR water after adjusting for the gestational week and other covariates. Since cardiovascular changes occur following eight weeks of gestation, we will include them in the analysis for pregnant women following eight weeks of gestation. We have clarified this in the revised manuscript.

Comment- Page 9, line 25 – please provide more information on the methods the trained promoter will use to encourage use of MAR. If the aim of the study is to evaluate uptake/use of the intervention, there needs to be discussion on the information campaigns and promotion of MAR water. Also, how will communities who are not selected to receive the intervention until later phases be motivated to continue participating in the study?

Responses- We have clarified the promoters' role in the revised manuscript. We will recruit one local community health promoter for each of the 16 MAR communities. The community health workers will visit households with promotional materials (e.g. flipcharts) to encourage household members to use MAR water for drinking and cooking purposes after roll out of intervention as per randomization. Prior to the stepped wedge trial, we conducted formative research among 18 pilot MAR communities to explore barriers and motivators for community MAR water use and system maintenance to inform a) behavior change promotional efforts for the stepped wedge trial study and b) system maintenance strategies. We conducted 24 in-depth interviews with adult household members that usually manage/collect drinking water or MAR water for family use and focus group discussions with the technical supervisors and caretakers of all 18 piloted MAR sites. From this study we found that demand for MAR water in some areas was sub-optimal, understanding of the impact of salinity on health was poor, and there were issues with perceptions of MAR water quality and availability. We

developed behavior change communications to address these barriers to MAR water uptake, used them to train staff to promote MAR water use and encouraged their use in promotion activities among communities comprising the stepped wedge trial.

Community health promoters in all sites will start promotion from the beginning of the study. They will describe adverse health consequences of drinking brackish water and potential health benefits of drinking low-salinity MAR water. As per the randomization schedule, they will inform the households when MAR water will be available and when to start consuming MAR water. Although infiltration of MAR water is ongoing for 1-2 years, communities will not have access to MAR water until implementers hand over the sites to community maintenance committees. Randomization was done prior commencement of the study and we will synchronize the community handover of MAR sites as per randomization.

As part of the MAR uptake evaluation, during follow-up visits the research staff will collect self-reported data on when each household starts using MAR water and whether households exclusively or intermittently use MAR water for drinking and cooking. They will also collect stored drinking and cooking water samples for measurement of electrical conductivity and ask the source of stored water.

Comment- GEE are sensitive to missing data. Please explain more about the multiple imputation methods you will consider and how you will evaluate if they are sufficient (page 14 line 46).

Response- We have revised the analysis plan and statistical method section in the revised manuscript. In the revised manuscript, we have proposed generalized linear mixed models with appropriate links instead of GEE for the primary analysis. We will consider random effects for community, households and participants, and a fixed effects of steps or visits for the primary analysis. Multiple imputations with chained equations will be used to jointly impute data on missing exposure, confounders, to preserve an unbiased association estimate if the data are missing at random.

Comment- Page 15, line 3 – please provide more details on the “random effect model.” This term can mean different things across disciplines. What sample size will be needed? Is it likely you will recruit enough pregnant women?

Response- We have revised the data analysis plan for the pregnant women. We will conduct generalized linear mixed models with appropriate links to assess whether access to water from MAR systems reduce blood pressure and urinary protein among the pregnant women considering random effects for each community and pregnant after adjusting for the gestational week and other covariates. We acknowledge that lack of sufficient pregnant women may mean that the analysis for this group is underpowered. We have clarified this in the revised manuscript.

Comment- Figure 2 – please list the sources of the data used in each map.

Response- We have listed the data sources to prepare both maps.

With these clarifications and suggestions noted above, I feel that publishing this protocol will make a good contribution to the literature and I look forward to reading the results of the study in the future.

Reviewer: 2

Reviewer Name: karla hemming

Institution and Country: Uni Bham uk

Please state any competing interests or state 'None declared': none

Please leave your comments for the authors below

Thank you for asking me to review this paper. It is well written but a few items need to be better reported,

these are as follows:

Comment- Will you adjust for time effects? This is crucial in this sort of study

Responses- Thanks for your comment. Yes, we will adjust for steps or visit number by considering the fixed effect of steps in generalized linear mixed models. We acknowledge that season may be a potential confounder affecting both water salinity and blood pressure.

Comment- When people are enrolled will they be told when their cluster will transition? What sort of information are they told when participants are enrolled into the study?

Responses- During the informed written consent process, we will inform each household that they will be randomly (a process like a lottery) selected when to start consuming MAR water as part of the scientific process of the study. Each MAR system will have a community management committee organized by the implementing partners. We will organize a meeting with implementers and inform them about the tentative date of allowing communities to consume MAR water based on randomization. The implementers will delegate the responsibilities of MAR systems to the community as per the randomization. Moreover, we will deploy local community health promoters who will conduct promotional visits and inform people when to start drinking MAR water as per randomization.

Comment- Sample size – please be clear that 1396 is the sample size needed under individual randomisation.

The sample size calculation is based on the Hussey and Hughes method which assumes different people are measured at each assessment time. If the TSS needed under individual randomisation is 1396×3 , then the number of people per cluster needed is $1296 \times 3 / 16$ which is in the region of 250 – these should be divided between the number of measurement points. The authors have applied a sample size formula for a cross-sectional design to a cohort design. Some acknowledgment of this is needed – it is likely that this means that authors calculation is conservative.

Response-Thanks for your comments on the sample size. As noted by the reviewer the sample size calculation was based on the formula proposed by Hussey and Hughes. ⁴ Although the formula assumed no within-participant correlation over time (cross-sectional design), we applied the formula for a cohort design. The salinity problem and the MAR water quality may be influenced by seasonality over the progression of the dry season and participants' blood pressure may respond differently to different level of sodium exposure in MAR drinking water. Therefore, it is important that we investigate the effect of the MAR system on the same participants over the entire dry season. Since we'll investigate the same participants in each step, we'll have less participant-level variability and sufficient power compared to a cross-sectional design to investigate the effect of MAR systems. We have clarified this in the revised manuscript.

For the sample size calculation, we considered a fixed cluster size of 60 participants ≥ 20 years old. We then followed the approaches of determining the number of clusters required.⁵ We agree with the reviewer that total sample size will be derived by multiplying the design effect with sample size needed under individual randomization. We also inflated the total sample size considering 10% loss to follow-up. We then calculated the total number of clusters required by dividing by the total steps and cluster size. We then calculated the number of steps that we need to randomize at each step by dividing total cluster by step. We calculated that 16 MAR communities will be required for the study with four communities randomized to access MAR water in each step.

Comment- A statistical methods section is needed. How the data will be analyzed must be given due consideration and be reported at this stage. Likewise, information on sub-group effects, sensitivity

analyses should also be reported.

Response- Thanks for the comment. We have revised the statistical method section in the revised manuscript. We have now proposed using generalized linear mixed models instead of GEE for the primary analysis. We have also reported the secondary and sub-group analyses.

Comment- This is a minor point, and almost obvious, but it is usually standard to report a justification for using a cluster design.

Response- Thanks for your point. We have included a justification of using cluster design in the revised manuscript. MAR system is a community-based intervention designed to supply low-salinity water for several households in a village or community. MAR intervention cannot be implemented and randomized at individual- or household-levels. Once a community will have access to MAR water, it is difficult to withhold the access of MAR water for some households. Therefore, we will conduct a cluster randomized trial where each community will be considered as a cluster.

Comment- I might have missed this, but a schematic representation of the design might be useful.

Response-We have added a schematic representation of the design in Figure 3.

Comment- Details on how the randomisation will be conducted, by whom, how it will be concealed from the clusters and when the clusters will be informed of their switch date. Who recruited the individual participants.

Response- We have included a detailed randomization process in the revised manuscript. Randomization was conducted by an investigator who was not directly involved in the implementation of the stepped wedge study. The study was not blinded as it is not possible to conceal access to MAR systems from the cluster participants. Although infiltration of rainwater and pond water into the brackish aquifer through MAR systems is ongoing for 1-2 years in each community, people do not have access to MAR water until a formal handover of MAR systems to communities. The implementers form a community management team who are responsible for maintenance of MAR systems following handover. We will synchronize the community handover with the randomization schedule in the 16 communities and our community health promoters will inform participants when the MAR water will be available for consumption. The study field research staff recruited the participants. The field research staff invited households who stated that they were interested in using MAR water from a list created by the implementers.

References:

1. Mente A, O'Donnell M, Rangarajan S, et al. Associations of urinary sodium excretion with cardiovascular events in individuals with and without hypertension: a pooled analysis of data from four studies. *The Lancet* 2016;388(10043):465-75.
2. UNICEF B. Bangladesh National Drinking Water Quality Survey of 2009 2009 [Available from: https://www.unicef.org/bangladesh/BNDWQS_2009_web.pdf].
3. Keswick BH, Satterwhite TK, Johnson PC, et al. Inactivation of Norwalk virus in drinking water by chlorine. *Appl Environ Microbiol* 1985;50(2):261-4.
4. Hussey MA, Hughes JP. Design and analysis of stepped wedge cluster randomized trials. *Contemporary clinical trials* 2007;28(2):182-91.
5. Hemming K, Taljaard M. Sample size calculations for stepped wedge and cluster randomised trials: a unified approach. *Journal of clinical epidemiology* 2016;69:137-46.

REVIEWER	Warren C. Jochem Dept. of Geography and Environment University of Southampton United Kingdom
REVIEW RETURNED	13-Jun-2017

GENERAL COMMENTS	Thank you for the opportunity to review this manuscript again. I appreciate that the authors have responded thoroughly to comments and revised this work. I have only minor notes for consideration. My earlier concern related to delaying the introduction of the MAR systems. It is now clear that the choice of the stepped-wedge design comes, in part, from how the implementers can hand over the systems to multiple communities. Synchronizing this study and randomizing that handover is an opportunity. I believe this point helps to justify the design and could be emphasized earlier in the introduction. I appreciate the added discussion of alternative study designs. The second paragraph of the Discussion section references the icddr,b's "IRB". Is this the Ethical Review Committee (ERC)? Strengths and limitations: The final bullet point makes reference to "individuals of different professions." Is compliance expected to vary by profession (which is not mentioned as a key variable in the study), or more generally by something like socioeconomic status? The manuscript would benefit from a review to correct minor grammar mistakes and to remove contractions.
---

REVIEWER	Karla Hemming University of Birmingham
REVIEW RETURNED	05-Jun-2017

GENERAL COMMENTS	Few minor comments:  1. When will you inform the villages of their cross-over date? 2. Was a "gatekeeper" identified for each village and their agreement sought for participation? 3. Do you take measurements which will not change at every measurement occasion (i.e. height?) 4. The focus on pregnant women seems too much. You are not powered for this analysis. I would suggest you simply say, pregnant women are not excluded and say why. You don't need to re-iterate this point continuously. 5. There may be a delay between receiving the intervention and seeing its affect. You could investigate this in a secondary analysis.
---

VERSION 2 – AUTHOR RESPONSE

Reviewer: 2

Reviewer Name: Karla Hemming

Institution and Country: University of Birmingham Please state any competing interests or state 'None declared': none

Please leave your comments for the authors below Few minor comments:

1. When will you inform the villages of their cross-over date?

Response: The promoters will inform the participants about the date of access to MAR water after forming the community management team for MAR systems hand over. We have clarified this in the "Randomization and intervention delivery" section of the revised manuscript.

2. Was a "gatekeeper" identified for each village and their agreement sought for participation?

Response: The implementers hire a caretaker or gatekeeper for each MAR community who is responsible for maintenance of the respective MAR system prior to the community hand over. The implementers form a community management team who are responsible for maintenance of each MAR system following handover. Site investigators will conduct several meeting with the gatekeepers to synchronize the community handover of MAR sites with the randomization schedule in the 16 communities. Agreement of the implementers and gatekeepers was sought at the beginning of the study during site selection for inclusion of any site in trial and access of MAR water as per the randomization. We have clarified this in the "Randomization and intervention delivery" section of the revised manuscript.

3. Do you take measurements which will not change at every measurement occasion (i.e. height?)

Response: We will take those measurements that may change in all visits. For instance, weight will be measured during all five visits, but height will be measured in one visit. We have revised the "Data collection methods" section accordingly.

4. The focus on pregnant women seems too much. You are not powered for this analysis. I would suggest you simply say, pregnant women are not excluded and say why. You don't need to re-iterate this point continuously.

Response- We agree with the reviewer that we are not powered for the pregnant women. In the trial we have enrolled only 17 pregnant women in 16 MAR communities. Although we previously intended to have separate analysis for the pregnant women, we currently plan to include pregnant women with the primary analysis of ≥ 20 years old participants. The "Statistical method" section has been revised to reflect this.

5. There may be a delay between receiving the intervention and seeing its affect. You could investigate this in a secondary analysis.

Response: Thanks for the suggestion. If the trial demonstrates drinking MAR water has health effects, we will investigate the time required for health effects following receiving MAR intervention in secondary analysis.

Reviewer: 1

Reviewer Name: Warren C. Jochem

Institution and Country: Dept. of Geography and Environment, University of Southampton, United Kingdom Please state any competing interests or state 'None declared': None declared

Please leave your comments for the authors below Thank you for the opportunity to review this

manuscript again. I appreciate that the authors have responded thoroughly to comments and revised this work. I have only minor notes for consideration.

My earlier concern related to delaying the introduction of the MAR systems. It is now clear that the choice of the stepped-wedge design comes, in part, from how the implementers can hand over the systems to multiple communities. Synchronizing this study and randomizing that handover is an opportunity. I believe this point helps to justify the design and could be emphasized earlier in the introduction.

I appreciate the added discussion of alternative study designs. The second paragraph of the Discussion section references the icddr,b's "IRB". Is this the Ethical Review Committee (ERC)?

Response: Thanks. Yes, icddr,b's IRB refers to Ethical Review Committee (ERC). We have replaced IRB by "Ethical Review Committee" to maintain consistency.

Strengths and limitations: The final bullet point makes reference to "individuals of different professions." Is compliance expected to vary by profession (which is not mentioned as a key variable in the study), or more generally by something like socioeconomic status?

Response: We agree with the reviewer that socioeconomic status is more appropriate than the profession. We have replaced profession by socioeconomic status.

The manuscript would benefit from a review to correct minor grammar mistakes and to remove contractions.

Response: We have removed the contractions in the revised manuscripts.

VERSION 3 – REVIEW

REVIEWER	Warren C. Jochem Department of Geography and Environment University of Southampton United Kingdom
REVIEW RETURNED	23-Jun-2017

GENERAL COMMENTS	Thank you to the authors for consideration of my comments and suggestions.
--

REVIEWER	Karla Hemming Institute of Applied Health Research, University of Birmingham
REVIEW RETURNED	10-Jul-2017

GENERAL COMMENTS	I am very happy with the revision and suggest it should be accepted without further revision.
---